# Temporal Bridges for Spatial Resolution: Enhancing Climate Data Super-Resolution with Bidirectional Alignment

## Abstract

High-resolution climate data is crucial for meteorological predictions and for informing decision support across diverse domains. However, the acquisition of such high-resolution climate information is often prohibitively costly, necessitating the development of data-driven meteorological prediction models. These models aim to generate fine-grained climate data from low-resolution inputs, a process termed climate data super-resolution (SR). Nevertheless, recent advancements in deep learning for climate data SR have primarily focused on leveraging single-frame spatial information, largely neglecting the temporal correlations between different time frames that could enhance SR outcomes. Furthermore, climate data are inherently stochastic and noisy, rendering widely used temporal alignment methods, such as optical flow models, ineffective in this context. Consequently, the development of a framework tailored for climate data SR that effectively captures implicit temporal correlations remains an unresolved challenge. To this end, we propose a novel Temporal-Enhanced framework with bidirectional temporal alignment. In essence, our framework establishes a temporal bridge to enhance spatial resolution in climate data SR through bidirectional alignment, leading to improved SR performance. Within this framework, Paired Latent Mapping achieves spatial alignment and noise reduction by unifying latent spaces. Then a Bidirectional Temporal Alignment captures temporal correlations by training forward and backward networks on consecutive latent frames. Temporal Enhanced Super-resolution then optimizes the entire framework for climate data SR. Experiments on large-scale real-world datasets demonstrated the superior performance of our framework.

## 1 Introduction

Climate data super-resolution (SR), also known as climate data downscaling, refers to transforming low-resolution climate data into a higher-resolution format, yielding more detailed and precise information for specific areas. Climate data SR is crucial for facilitating decision-making and enabling high-resolution meteorological predictions. The high-resolution climate information is essential for decision support in many sectors such as urban management, transportation optimization, and disaster prevention Lam et al. (2023); Hertwig et al. (2021). However, deploying sufficient sensors to collect such high-resolution climate information is often impractical due to cost and infrastructure barriers.

To address the scarcity of observational data, a promising and cost-effective approach is to leverage the abundance of historical climate data by employing data-driven meteorological prediction models to generate fine-grained (i.e., higher-resolution) climate data from low-resolution inputs. While traditional numerical simulations and statistical interpolation have been extensively explored for climate data SR, these methods can be computationally intensive or may neglect essential spatial correlations Lin et al. (2023). Recent advancements in models like ForecastNet Pathak et al. (2022), Pangu-Weather Bi et al. (2023), and GraphCast Lam et al. (2023) demonstrate the feasibility and potential of training foundational models for meteorological prediction using large-scale climate datasets and transformer-based architectures. However, existing data-driven models, including those mentioned above, typically predict at coarse resolutions (e.g., $30 \times 30$ km), which are insufficient

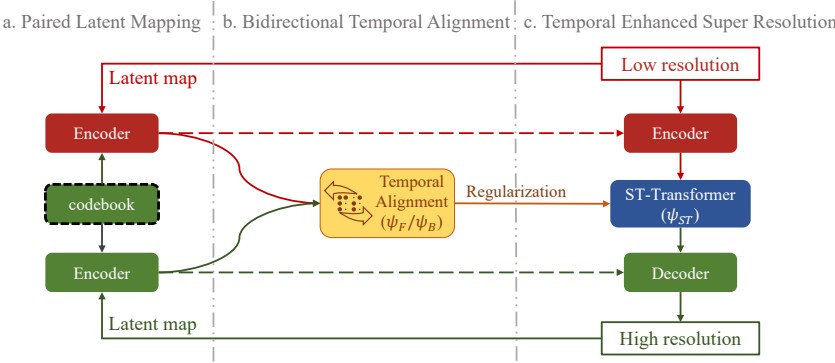

Figure 1: Overview of the proposed framework. a. Paired Latent Mapping module maps low and high resolution climate data to unified latent space using a shared codebook. b. Bidirectional Temporal Alignment module trains temporal alignment network as the regularization term of super-resolution training. c. Temporal Enhanced Super-resolution module conduct super-resolution training based on a.and b.

for the decision-making needs of sectors such as urban management and disaster prevention, as previously discussed. Therefore, the development of effective super-resolution methods specifically tailored for climate data SR is critically important.

While deep learning approaches have been applied to climate data SR, our insight is that the inherent temporal correlations within climate data across different time steps have been largely overlooked. Deep learning models such as Convolutional Neural Networks (CNNs) Liu et al. (2020); Lin et al. (2023); Kheir et al. (2023) and Generative Adversarial Networks (GANs) Stengel et al. (2020) have shown promise in climate data SR, and the recently developed ClimaX model, pre-trained on climate data and fine-tuned for SR, has achieved state-of-the-art results Nguyen et al. (2023). However, existing research has primarily focused on exploiting spatial information within individual time frames, neglecting inter-time correlations Nguyen et al. (2023). Given that climate is a continuously evolving phenomenon driven by complex interactions between atmospheric conditions and Earth's physical features, and considering the significant temporal correlations exhibited by variables like temperature and wind speed, we hypothesize that incorporating these temporal relationships, in conjunction with spatial information, can enhance high-resolution outputs and improve SR performance.

While the incorporation of temporal relationships has been a common practice in video super-resolution (VSR) Tu et al. (2022), effectively applying these techniques to climate data presents significant challenges. The general idea of VSR for leveraging temporal information involves utilizing optical flow models to estimate motion between consecutive frames Tu et al. (2022); Bari et al. (2023). However, a fundamental assumption of these optical flow models is that pixel brightness remains relatively consistent over time, which enables the estimation of motion by tracking pixel intensity changes across frames Bari et al. (2023). In contrast, this assumption is invalid for climate data which exhibit inherently stochastic and prone to sudden changes Palmer (2019). Although global and regional climate models can partially capture temporal variations in climate data Lynch (2008); Palmer (2019), these model-regulated hidden temporal correlations cannot be explicitly observed or annotated like the motion of objects. Moreover, observational limitations and instrumental inaccuracies often introduce substantial spatial noise into climate data Privé et al. (2013), which can significantly hinder the performance of super-resolution tasks. These unique challenges underscore the need for a tailored framework that can effectively capture the implicit temporal correlations in climate data while mitigating the inherent noise characteristics of such data.

To address the research gap in climate data super-resolution (SR), we propose a novel Temporal Enhanced framework with bidirectional temporal alignment. This framework explicitly models the bidirectional temporal correlations inherent in climate data using temporal alignment networks to achieve enhanced SR performance. As illustrated in Figure 1, the framework comprises three key modules: (a) the Paired Latent Mapping module, (b) the Bidirectional Temporal Alignment module, and (c) the Temporal Enhanced Super-resolution module. The Paired Latent Mapping module (Figure 1a) is specifically designed to mitigate the inherent spatial noise present in climate datasets.

Leveraging Vector Quantised-Variational Autoencoders (VQ-VAE) Van Den Oord et al. (2017), this module maps climate data into a unified latent space, effectively reducing noise and establishing a robust spatial foundation for subsequent SR tasks. To ensure representational consistency across resolutions, VQ-VAE models for both low and high-resolution data are paired using a shared codebook. Building upon this spatial foundation, the Bidirectional Temporal Alignment Module (Figure 1b) is designed to achieve temporal alignment in climate data SR. By exploring the relationships between climate data across different time frames, this module trains two distinct alignment networks – forward and backward – to effectively leverage bidirectional temporal dependencies. Both alignment networks are then integrated as the regularization term, thereby enhancing the detail and accuracy of the reconstructed high-resolution outputs. Finally, building on the above two modules, the Temporal Enhanced Super-resolution module (Figure 1c) trains the SR process in two steps. Firstly, the module focuses on SR within the latent space, incorporating the bidirectional temporal alignment networks as regularization. Secondly, to refine the results in the original climate data domain, we unfreeze and fine-tune the encoder and decoder components of the latent mapping.

Following the settings of previous studies Nguyen et al. (2023), we conduct extensive experiments on a large dataset constructed with Coupled Model Intercomparison Project Phase 6 (CMIP6) O'Neill et al. (2016) and European Centre for Medium-Range Weather Forecasts Reanalysis 5 (ERA5) Hersbach et al. (2020) data and achieve state-of-the-art results. Our contributions are summarized as follows:

- We propose a carefully designed Temporal Enhanced SR framework tailored for the climate data SR task which aims to significantly boost SR performance.

- We develop a novel temporal alignment technique that can adapt to the stochastic nature of climate data and effectively leverages temporal information to enhance spatial details, establishing Temporal Bridges for Spatial Resolution in climate data SR.

- We conduct thorough evaluations on CMIP6 and ERA5 datasets, which demonstrates the superior performance of our framework in real-world datasets.

## 2 RELATED WORK

### 2.1 CLIMATE DATA SUPER-RESOLUTION

Traditional climate data super-resolution techniques can be categorized into dynamic approaches and statistical approaches. Dynamic methods primarily relies on regional climate models (RCMs), while statistical methods improves resolution by establishing statistical relationships between GCM outputs and ground-based observational data, such as Perfect Prognosis (PP) Landman et al. (2001), Model Output Statistics (MOS) Eden & Widmann (2014), and weather generators Wilks (2010). Recently, there has been a notable increase in the application of deep learning methods for climate data SR. Techniques such as CNNs Liu et al. (2020); Lin et al. (2023); Kheir et al. (2023), GANs Stengel et al. (2020), and Diffusion Models Aich et al. (2024); Watt & Mansfield (2024) are significantly explored to handle the climate data. The ClimaX Nguyen et al. (2023) model has achieved good performance in various downstream tasks, including SR, through deep pre-training on climate data.

### 2.2 TEMPORAL ALIGNMENT IN SUPER-RESOLUTION

From the perspective of technique, temporal alignment is critical due to the high correlation and spatial displacement between adjacent frames in the super-resolution domain. Traditional methods for temporal alignment typically use optical flow estimation methods based on gradients or block matching Caballero et al. (2017); Liu et al. (2017); Tao et al. (2017), which have been widely applied in the scenario of VSR. Recent advancements have seen the emergence of deep learning-based optical flow estimation, which offers more accurate and robust alignment Ranjan & Black (2017). More recently, modern SR works have begun to integrate these sophisticated optical flow models by utilizing pre-trained networks to significantly enhance performance Xue et al. (2019); Chan et al. (2022); Liang et al. (2024; 2022). In particular, VRT Liang et al. (2024) achieves state-of-the-art results on various VSR benchmarks. Despite these advances, there is currently no research that applies temporal alignment techniques to the field of climate data SR.

# 3 METHOD

Figure 2: Architecture of the proposed framework. a. Low and high-resolution climate data are mapped to latent space using VQ-VAEs with a shared codebook. b. Temporal alignment network is pre-trained by exploring correlations of data across time frames. c. Super-resolution training are conducted initially in latent space and then in climate domain.

## 3.1 PROBLEM DESCRIPTION

The primary objective of climate data SR is to transform low-resolution (LR) input frames, denoted as $I_{LR} \in \mathbb{R}^{T \times V \times H \times W}$, into high-resolution (HR) output frames, denoted as $I_{HR} \in \mathbb{R}^{T \times V \times sH \times sW}$. Here, $T$ denotes the size of the temporal window, $V$ denotes the number of climate variables (e.g. temperature), $s$ denotes the upsampling factor, and $H$ and $W$ denote the horizontal and vertical dimensions of the Earth's grid format respectively.

## 3.2 MODEL FRAMEWORK OVERVIEW

The proposed Temporal Enhanced SR framework consists of three components: the Paired Latent Mapping module, the Bidirectional Temporal Alignment module, and the Temporal Enhanced Super-resolution module (Figure 2). The Paired Latent Mapping module (Figure 2a) maps both high and low-resolution climate data into a unified latent space to reduce the impact of noise in the spatial dimension. The Bidirectional Temporal Alignment module (Figure 2b) leverages relationships between climate data across different time frames by training two temporal alignment networks, which are integrated into the SR loss function to utilize temporal information. The Temporal Enhanced Super-resolution module (Figure 2c) trains climate data based on the two modules above, which involves a two-step process, initially focusing on latent space representations and subsequently on the climate data domain itself. In the subsequent sections, we provide a comprehensive introduction and discussion regarding these three components.

## 3.3 PAIRED LATENT MAPPING

To cope with the noise issue in climate data, we develop a Paired Latent Mapping module that maps both high and low-resolution data into a unified and discrete latent space using Vector Quantised-Variational Autoencoder (VQ-VAE) (Figure 2a). The encoder in VQ-VAE generates discrete vectors, drawn directly from a learnable, fixed codebook, which enhances the model's stability and interpretability. Moreover, we introduce a paired latent mapping architecture in which both the low and high resolution VQ-VAE models share the same codebook. This paired design ensures uniform feature representation across different spatial resolution, facilitating improved performance in SR

applications. The latent mapping module is formally defined as:

$$Z_{LR}(t) = q(\phi_{LR}(I_{LR}(t))), \quad \hat{I}_{LR}(t) = \omega_{LR}(Z_{LR}(t))$$
$$Z_{HR}(t) = q(\phi_{HQ}(I_{HR}(t))), \quad \hat{I}_{HR}(t) = \omega_{HR}(Z_{HR}(t))$$

(1)

where $Z_{LR}(t)$ and $Z_{HR}(t)$ denote the latent representations for low and high-resolution data inputs $I_{LR}(t)$ and $I_{HR}(t)$ at time $t$, respectively. The mappings $\phi_{LR}$ and $\phi_{HR}$ are encoder functions specifically designed for different resolutions, while $\omega_{LR}$ and $\omega_{HR}$ are decoder functions. The function $q(\cdot)$ represents the vector quantization operation, which discretizes the continuous latent outputs into a shared codebook. $\hat{I}_{LR}(t)$ and $\hat{I}_{HR}(t)$ are the reconstructed low and high-resolution data outputs from their respective latent representations at time $t$.

During the training process, we start by training a VQ-VAE model on high-resolution climate data. The encoder incorporates six layers of residual convolutional networks and four Transformer network layers to effectively capture spatial correlations. The decoder uses upsampling layers that mirror the encoder's structure, progressively restoring data resolution. This architecture is then replicated for low-resolution data training, using the same codebook trained from the high-resolution data, with the codebook's parameters frozen to ensure feature space consistency across different resolutions. Both Mean Squared Error (MSE) and commitment loss are used to measure reconstruction quality and maintain encoding vector consistency.

### 3.4 BIDIRECTIONAL TEMPORAL ALIGNMENT

After the data is mapped to the latent space, we develop a Bidirectional Temporal Alignment module to leverage temporal information in climate data SR (Figure 2b). Consider the latent space representations at different resolutions at time $t$, denoted $Z_{LR}(t)$ and $Z_{HR}(t)$. Drawing on concepts in VSR, the temporal correlation between consecutive low-resolution time frames, $Z_{LR}(t-1)$ and $Z_{LR}(t)$, shares common features with the correlation in high-resolution frames $Z_{HR}(t-1)$ and $Z_{HR}(t)$. Therefore, including $Z_{LR}(t-1)$ and $Z_{LR}(t)$ can assist in the mapping from $Z_{HR}(t-1)$ to $Z_{HR}(t)$. If a network $\psi_F$ is trained using $Z_{LR}(t-1)$, $Z_{LR}(t)$, and $Z_{HR}(t-1)$ to predict $Z_{HR}(t)$, it is reasonable to assume that $\psi_F$ captures some shared temporal correlation features across different resolutions, which can then be used in temporal alignment (forward alignment). Correspondingly, training another network $\psi_B$ with inputs $Z_{LR}(t+1)$, $Z_{LR}(t)$, and $Z_{HR}(t+1)$, aiming to predict $Z_{HR}(t)$, can also be used for temporal alignment (backward alignment). Together, $\psi_F$ and $\psi_B$ form the core of the Bidirectional Temporal Alignment Module. They are formally defined as:

$$\check{Z}_{HR}(t) = \psi_F(Z_{LR}(t-1), Z_{HR}(t-1), Z_{LR}(t)),$$
$$\check{Z}_{HR}(t) = \psi_B(Z_{LR}(t+1), Z_{HR}(t+1), Z_{LR}(t)),$$

(2)

Here, both forward and backward alignments are incorporated, drawing from the settings of optical flow models commonly used in VSR.

During the training process, the forward alignment network $\psi_F$ initially concatenates inputs $Z_{LR}(t-1)$, $Z_{LR}(t)$, and $Z_{HR}(t-1)$. This combined input is processed through two Transformer blocks, followed by a convolutional block to refine the features, and then through two additional Transformer layers to predict $Z_{HR}(t)$. The network is trained using Mean Squared Error (MSE) to minimize the differences between predicted and high-resolution features. Similarly, the backward alignment network $\psi_B$ concatenates inputs $Z_{LR}(t+1)$, $Z_{LR}(t)$, and $Z_{HR}(t+1)$ and follows the same architectural sequence to predict $Z_{HR}(t)$.

### 3.5 TEMPORAL ENHANCED SUPER RESOLUTION

Building on the latent mapping and temporal alignment network, we conduct Temporal Enhanced SR training (Figure 2c). This training process includes two steps: the first focuses on the latent space, and the second on the climate data domain itself.

Step 1: This step focuses on latent space representations at different resolutions which are represented as $Z_{LR}(t)$ and $Z_{HR}(t)$. The spatial-temporal SR network $\psi_{ST}$ is responsible for transforming $Z_{LR}(t)$ into $Z_{HR}(t)$ as:

$$\hat{Z}_{HR}(t) = \psi_{ST}(Z_{LR}(t))$$

(3)

As observed in VSR, directly minimizing using Mean Squared Error (MSE) loss $\min_{\psi_{ST}} \text{MSE}(\psi_{ST}(Z_{LR}(t)), Z_{HR}(t))$ may not effectively leverage the temporal correlation between time frames. To enhance the model's ability to utilize these temporal correlations, we integrate the temporal alignment networks $\psi_F$ and $\psi_B$ defined in Section 3.4 into the loss function as regularization components:

$$
\begin{aligned}
\mathcal{L} = \frac{1}{T} \sum_{t=0}^{T-1} \Big[ &\text{MSE}(\psi_{ST}(Z_{LR}(t)), Z_{HR}(t)) \\
&+ \cdot \text{MSE}(\psi_F(Z_{LR}(t-1), Z_{LR}(t), \psi_{ST}(Z_{LR}(t-1))), Z_{HR}(t)) \\
&+ \cdot \text{MSE}(\psi_B(Z_{LR}(t+1), Z_{LR}(t), \psi_{ST}(Z_{LR}(t+1))), Z_{HR}(t)) \Big]
\end{aligned}
\tag{4}
$$

Note that the alignment network in Equation 4 is different from Equation 2. For instance, the high resolution input $Z_{HR}(t-1)$ in $\psi_F$ is replaced by $\psi_{ST}(Z_{LR}(t-1))$. This is to comply with the specifications of super-resolution tasks, which exclusively use low-resolution data as inputs.

During the training process, the $\psi_{ST}$ first applies patch embedding and positional encoding to $Z_{LR}(t-1)$, $Z_{LR}(t)$, and $Z_{LR}(t+1)$. These encoded inputs are then processed through two Transformer modules and a 3D transposed convolution layer, producing the predicted values $\hat{Z}_{HR}(t-1)$, $\hat{Z}_{HR}(t)$, and $\hat{Z}_{HR}(t+1)$ for $Z_{HR}(t-1)$, $Z_{HR}(t)$, and $Z_{HR}(t+1)$. By using the $\psi_F$, and $\psi_B$ networks with inputs $Z_{LR}(t-1)$, $Z_{LR}(t)$, $\hat{Z}_{HR}(t+1)$, and $\hat{Z}_{HR}(t-1)$, the training for $Z_{HR}(t)$ can be conducted. The $\psi_{ST}$ employs Mean Squared Error (MSE) to minimize the differences between all predicted values and the actual high-resolution features.

Step 2: After training the network for a specified number of epochs, we unfreeze the encoder and decoder components. The regularization term from the alignment network is retained to maintain temporal information. We then shift our optimization focus to fine-tuning the network's output using the Latitude-weighted Mean Squared Error (LMSE) as defined in Rasp et al. (2020). This specialized loss function, detailed below, incorporates a latitude weighting factor to account for the varying area sizes at different latitudes on a global grid as:

$$
\mathcal{L}_{\text{LMSE}} = \sum_{i=0}^{T-1} \sum_{v=0}^{V-1} \sum_{h=0}^{sH-1} \sum_{w=0}^{sW-1} \frac{L(h) \left( \hat{I}_{HR}^{v,h,w}(t) - I_{HR}^{v,h,w}(t) \right)}{T \times V \times sH \times sW},
\tag{5}
$$

in which $L(h)$ is the latitude weighting factor:

$$
L(h) = \frac{\cos(\text{lat}(h))}{\frac{1}{H} \sum_{h'=0}^{H-1} \cos(\text{lat}(h'))}
\tag{6}
$$

Here, $\text{lat}(h)$ indicates the latitude corresponding to the $h$-th row in the grid. This weighting method adjusts for the unequal area representation across different latitudes, enhancing the model's accuracy in geographic areas.

### 3.6 INFERENCE

During the inference phase, the trained low resolution encoder $\phi_{LR}$, high resolution decoder $\omega_{HR}$, along with the SR network $\psi_{ST}$, are employed to transform low-resolution climate data into high-resolution data. The inference process is defined as:

$$
\begin{aligned}
Z_{LR}(t) &= q(\phi_{LR}(I_{LR}(t))) \\
\hat{Z}_{HR}(t) &= \psi_{ST}(Z_{LR}(t)) \\
\hat{I}_{HR}(t) &= \omega_{HR}(Z_{HR}(t))
\end{aligned}
\tag{7}
$$

## 4 EXPERIMENT

In this section, we conduct extensive experiments on real-world datasets to evaluate our proposed framework for climate data SR. First, we compare the super-resolving performance on five climate

Table 1: Performance of Our Method and the baselines on the SR task from CMIP6 (5.625°) to ERA5 (1.40625°). The mean and standard deviation are obtained through five random runs.

| Method | Features (RMSE) ↓ | | | | |
|---|---|---|---|---|---|
| | Z500 | T850 | T2m | U10 | V10 |
| VRT | 1098.95(9.94) | 5.58(0.12) | 6.33(0.18) | 4.24(0.01) | 4.24(0.01) |
| SwinIR | 1099.07(3.41) | 5.54(0.04) | 6.22(0.12) | 4.23(0.01) | 4.24(0.02) |
| ClimaX | 1088.42(2.00) | 5.51(0.01) | 6.11(0.02) | 4.23(0.01) | 4.24(0.01) |
| Our Method | **1077.73(1.02)** | **5.41(0.01)** | **6.02(0.01)** | **4.22(0.01)** | **4.23(0.01)** |

features between our approach and state-of-the-art methods for climate SR tasks (Section 4.2). In addition, we conduct an ablation study to verify the effectiveness of each design in our model (Section 4.3), and also investigate the efficiency of our method (Section 4.4).

## 4.1 EXPERIMENTAL SETUP

**Dataset.** Following Nguyen et al. (2023), we construct the real-world climate SR dataset based on CMIP6 data and ERA5 data, where CMIP6 and ERA5 provide the low-resolution and high-resolution data, respectively. We next introduce how to construct the dataset for performance evaluation from these two data sources.

ERA5 is a reanalysis data for global climate in the past decades Hersbach et al. (2020). The data used in this study is derived from WeatherBench Rasp et al. (2020) which provides data in three resolutions of 5.625°, 2.8125°, and 1.40625° for the period from 1979 to 2018, serving as a benchmarking framework to facilitate comparisons of data-driven approaches in weather forecasting.

In accordance with ClimaX Nguyen et al. (2023), we use the data at 1.40625° resolution as the high resolution target in our experiments. For detailed characteristics of the raw ERA5 data, please refer to the ECMWF documentation available online Hersbach et al. (2020).

CMIP6 is a global project that provides climate model data covering historical and future periods. We derive the six-hour interval, 5.625° resolution data through the official CMIP6 search interface at CMIP6 Data Portal O'Neill et al. (2016). For detailed dataset settings, refer to Appendix as the low resolution input.

To construct the SR dataset, we further align the sampling interval and the time span of two sources of data. For ERA5 data, which originally collects data every hour, we perform downsampling to match the six-hour sampling interval of CMIP6; while for CMIP6, we use data from after 1979 to match the time span of ERA5. In this way, we obtain the super-resolution dataset spanning 37 years (1979-2015) with a six-hour sampling interval. We split the data into three sets, in which the training data is from 1979 to 2010, the validation data is in 2011 and 2012, and the test data is from 2013 and 2015. For evaluating the SR performance of all comparing methods, we follow ClimaX Nguyen et al. (2023) and select the same five key climate variables, which are geopotential at 500 mb (Z500), temperature at 850 mb (T850), 2-meter temperature (T2m), 10-meter u-component of wind (U10), and 10-meter v-component of wind (V10).

**Baseline.** We compare our framework with several state-of-the-art baseline, including a climate SR model ClimaX Nguyen et al. (2023), and two video SR approaches VRT Liang et al. (2024) and SwinIR Liang et al. (2021). VRT includes an optical flow to consider the temporal correlation. For SwinIR, we adapt it by replacing the internal Swin Transformer Liu et al. (2021) with a Video Swin Transformer Liu et al. (2022) to better suit our task.

**Metric.** We evaluate all methods using Latitude-weighted Root Mean Square Error (RMSE), which was commonly used in existing works Vandal et al. (2017); Liu et al. (2020). It is calculated by:

$$\text{RMSE} = \frac{1}{N} \sum_{k=0}^{N-1} \sqrt{\frac{1}{H \times W} \sum_{h=0}^{H} \sum_{w=0}^{W} L(h)(\hat{I}_{HQ}^{h,w} - I_{HQ}^{h,w})}, \tag{8}$$

where $L(h)$ is the weighting factor defined in Equation 6.

**Implementation.** We implement the comparing methods based on the configurations described in

the ClimaX paper Nguyen et al. (2023) and other established methods Liang et al. (2024; 2021). For ClimaX model, we uses all climate variables at a single time point as input, denoted by $I_{LQ}^v(t)$, where $v \in \{0, 2, \ldots, V-1\}$, and outputs five variables at the same time point, denoted by $I_{HQ}^v(t_0)$ for $v \in \{0, 1, 2, 3, 4\}$. While for our proposed framework and other methods that utilize temporal information, they take four consecutive time points, each with five features, as input, denoted as $I_{LQ}^v(t)$ for $t \in \{0, 1, 2, 3\}$ and $v \in \{0, 1, 2, 3, 4\}$, and produce the corresponding high-resolution features at each time point, denoted as $I_{HQ}^v(t)$ for $t \in \{0, 1, 2, 3\}$ and $v \in \{0, 1, 2, 3, 4\}$.

Next, we introduce other hyper-parameter settings in our framework. For the training of VQ-VAE in latent mapping, we set the learning rate to $4.5e-5$, the batch size to 24, and train the model for a maximum of 50 epochs using the Adam optimizer to ensure stable training and effective convergence. For the temporal alignment network training, we also set the learning rate at $4.5e-5$, but reduce the batch size to 5 due to memory constraints, with training still conducted for 50 epochs using the Adam optimizer. In the SR training, we use a much lower learning rate of $1e-6$ to facilitate fine-tuning and used a batch size of 1 to manage high memory usage, while maintaining the same optimizer and epoch count.

**Computational infrastructure.** In our experiments, we conduct comparison and ablation experiments using a computing platform equipped with four NVIDIA V100 GPUs.

## 4.2 SUPER-RESOLUTION PERFORMANCE EVALUATION

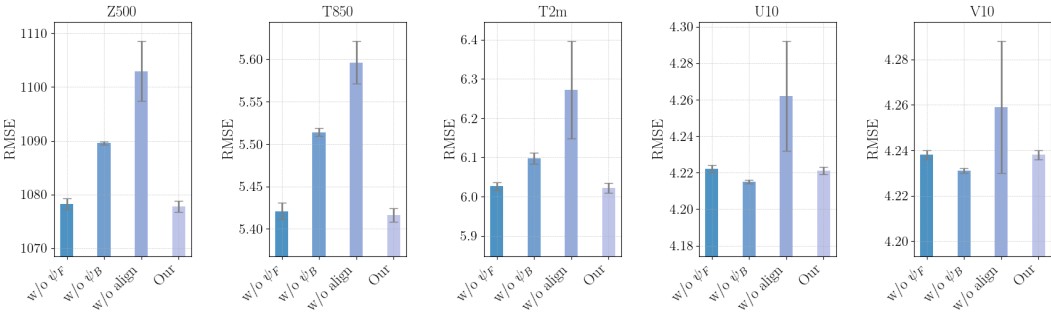

Figure 3: Ablation study.

To evaluate the impact of temporal alignment networks on model performance, we conduct an ablation study comparing our method with models under the following configurations:

- **w/o $\psi_F$**: removing the forward alignment network and keeping the rest.
- **w/o $\psi_B$**: removing the backward alignment network and keeping the rest.
- **w/o align**: removing all temporal alignment networks.

Table 1 compares the performance of our method with three baselines (VRT, SwinIR, and ClimaX) in super-resolving five climate variables from CMIP6 (5.625°) to ERA5 (1.40625°). The mean and standard deviation of RMSE defined in Equation 8 are reported over five random runs. It is observed that our framework achieves the lowest RMSE across all variables–Z500 (1077.73), T850 (5.41), T2m (6.02), U10 (4.22), and V10 (4.23)–significantly outperforming all baselines, revealing the superior performance of our method (see visualization examples in Appendix). As the second-best model, ClimaX shows better performance than VRT and SwinIR with RMSE values: Z500 (1088.42), T850 (5.51), T2m (6.11), U10 (4.23), and V10 (4.24). This improved performance is likely due to ClimaX using more climate variables as input, enhancing the SR performance. However, even with ClimaX utilizing more input variables, our method still outperforms it, notably reducing the RMSE of Z500 from 1088.42 (2.00) to 1077.73 (1.02) and T2m from 6.11 (0.02) to 6.02 (0.01). It is also worth noting that the performances of VRT and SwinIR are quite close, despite VRT's use of optical flow models. Specifically, VRT records RMSE values of Z500 (1098.95), T850 (5.58), T2m (6.33), U10 (4.24), and V10 (4.24), while SwinIR, which does not use optical flow, records RMSE values of Z500 (1099.07), T850 (5.54), T2m (6.22), U10 (4.23), and V10

(4.24). This similarity indicates that optical flow models, despite their utility in video data for handling visual continuity and motion, may not effectively address the spatial and temporal variability in climate data.

In contrast, our method incorporates a specifically designed temporal alignment mechanism tailored to address the inherent characteristics of climate data. This design enables our method to outperform both VRT and SwinIR under the same input-output conditions, and even exceed the results of ClimaX, despite the latter having more input variables.

### 4.3 ABLATION STUDY

As depicted in Figure 3, the bidirectional temporal alignment we designed significantly contributes to the super-resolution (SR) results. When $\psi_F$ and $\psi_B$ are removed, there is a substantial decline in model performance across all five variables. Meanwhile, the contributions of alignment in different directions vary among the variables. Compared to U10 and V10, the backward alignment appears to be more critical for Z500, T850, and T2m, as the removal of $\psi_B$ leads to greater drops in SR performances of these variables.

### 4.4 EFFICIENCY EVALUATION

Table 2: Training and Inference Efficiency Comparison

| Model | Training Time (h/epoch) | Inference Time (s/instance) | Model Size (MBytes) |
|---|---|---|---|
| VRT | 0.598 | 0.058 | 30.7 |
| SwinIR | 0.130 | 0.016 | 25.1 |
| ClimaX | 0.423 | 0.081 | 110 |
| Our Model | 0.355 | 0.05 | 180 |

To evaluate the applicability of our model, we compare it with existing techniques including VRT, SwinIR, and ClimaX on a computing platform equipped with four NVIDIA V100 GPUs. Table 2 lists the training time, inference time, and model size for each model. Despite its larger size of 180 MBytes compared to VRT's 30.7 MBytes and ClimaX's 110 MBytes, our model maintains superior efficiency. It achieves a faster training time of 0.355 hours per epoch, outperforming VRT at 0.598 hours and ClimaX at 0.423 hours. Furthermore, it provides faster inference, taking 0.05 seconds per global instance (i.e. the global climate data for each time frame), which is much quicker than 0.081 seconds of ClimaX and 0.058 seconds of VRT. The results reveal our model's ability to achieve a balance between efficiency and high performance, demonstrating stronger practicality in real-world applications compared to others.

## 5 CONCLUSION

In this study, we develop a novel Temporal-Enhanced framework that utilizes temporal information to enhance SR performance. The core of the framework involves training two temporal alignment networks for SR task to achieve temporal alignment. This design effectively addresses the challenge of stochastic variations in climate data, which hinder the use of conventional optical flow methods in VSR. Experiments on CMIP6 and ERA5 datasets confirm the superior performance of our framework. However, the applicability of our framework is not limited to these datasets. Future studies could extend our proposed framework to similar climate datasets. Additionally, the proposed temporal alignment architecture has the potential to be used for the super-resolution of spatial-temporal data in other fields. Despite significant achievements, our approach has limitations. We followed the practice in optical flow models, which mainly model the mapping relationships between adjacent time frames. As a result, our framework does not consider the temporal correlation between longer time frames. Future research could explore ways to integrate these longer temporal correlations to investigate whether they enhance the performance in climate data SR.

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

## .1 DETAILS FOR CMIP6 DATA

Following the configurations in ClimaX Nguyen et al. (2023), the criteria for data selection include:

- **Experiment ID:** 'historical' — focusing on historically simulated climate conditions.
- **Table ID:** '6hrPlevPt' — which indicates data sampled every six hours at specific pressure levels.
- **Variant Label:** 'r1i1p1f1' — a code identifying simulations differentiated by initial conditions ('r'), initialization methods ('i'), physics adjustments ('p'), and forcing variants ('f').

With the selected data, we regrid them to 5.625° resolution, which will be used as the low-resolution input to perform the super-resolution task. This regriding process is implemented by *xesmf* Python package with bilinear interpolation.

## .2 VISUALIZATION OF SUPER-RESOLUTION RESULTS

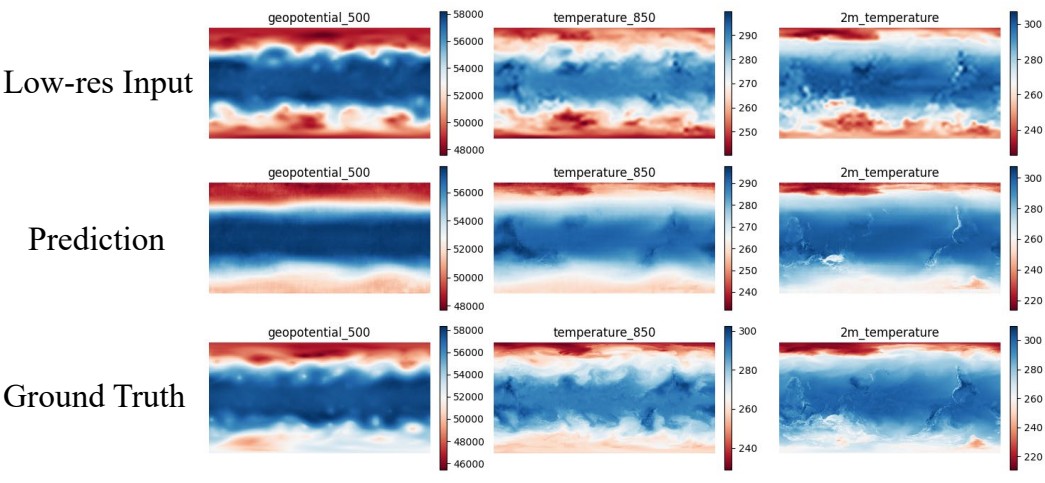

Figure 4: Examples of Model Output.

Figure 4 presents comparisons between low resolution inputs (top), model predictions (middle), and the ground truth (bottom) in our experiment.

