# OpenReview forum: "Temporal Bridges for Spatial Resolution: Enhancing Climate Data Super-Resolution with Bidirectional Alignment"
_ICLR.cc/2026/Conference — Submitted to ICLR 2026_

### Official Review · Reviewer_3yq4 · 2025-10-29

**Soundness:** 2
**Presentation:** 3
**Contribution:** 2
**Rating:** 4
**Confidence:** 4

**Summary:**

The authors develop a framework to address super resolution tasks on weather data. This is done through an architecture with three components: a VQ-VAE codebook that embeds the high and low resolution data, a bidirectional temporal alignment, and a temporal enhancement model. The authors draw inspiration from video super resolution to utilize information from previous times to facilitate the super resolution task.

**Strengths:**

The authors develop a novel method to add temporal information to help improve super resolution. Compared to their benchmarks, their method outperforms.

**Weaknesses:**

I am not convinced that this work offers sufficient improvement over more recent methods. This model also seems to be fairly complex, without justifying the complexity. A more detailed ablation study would be appreciated. For example, why is VQVAE the best option?

**Questions:**

- One significant limitation seems to be the availability of high resolution data used to train the codebook. How can this be offset?

- While $\psi_B$ is shown to assist the SR training, can the authors provide some understanding of how using future information to predict past information will not cause any issues. Physical systems evolve such that entropy (disorder) continuously increases. Therefore, going forward and backward in time is not a symmetric process. Granted, such arguments may not be too important for the timescale the authors are looking at. Still, some intuition of how $\psi_B$ may negatively affect the model is helpful.

- On line 270 the authors state (without citation) that minimizing VSR with only MSE is not optimal. This seems to be a key concept as the loss with $\psi_B$ and $\psi_F$ directly follows from this logic. There is a lack of reasoning and evidence to justify this loss. While the loss may be justifiable, the authors colloquially say in 270 that MSE alone is not good, with no justification or citations, and expect the reader to simply believe them. This is lacks sufficient scientific rigor. Please add citations, explanations, and some intuition to back up this claim which is important to why you need this loss and consequently $\psi_B$ and $\psi_F$.

- The benchmark models are from 2021, 2023, and 2024, but none from the first half of 2025, nor more from 2024. Can the authors either add more benchmarks or explain why they chose not to include more modern models.

- I do not see a comparison with diffusion or flow-based models, which have been shown to improve super-resolution. The authors state that VRT includes an optical flow, but it is unclear if this is a generative ML method like flow-based models. A good (and recent) comparison for this work would be "FLEX: A Backbone for Diffusion-Based Modeling of Spatio-temporal Physical Systems" which also uses temporal information to improve super resolution tasks, but uses a diffusion model framework.

- If the goal is to resolve weather data to the same resolution as data that currently exists, then why not train at that resolution and supplement the training with lower resolution data? Can the authors explain why it is better to apply their method rather than multi-resolution training. Do the authors expect that this method can be applied to the highest resolution data available and improve that resolution beyond what it was trained on? Currently, that is unclear as the supervised loss uses high-resolution data.

---

### Official Review · Reviewer_zjLD · 2025-10-30

**Soundness:** 2
**Presentation:** 3
**Contribution:** 2
**Rating:** 4
**Confidence:** 3

**Summary:**

This paper proposes a new method for generating high-resolution climate data from low-resolution inputs, i.e., climate data super-resolution (SR). The authors argue that existing works neglect temporal correlations between time frames and struggle with the stochastic and noisy nature of climate data. To address these issues, the authors propose a novel Temporal-Enhanced framework with bidirectional temporal alignment. The framework consists of three modules: (1) Paired Latent Mapping, which maps both low- and high-resolution data into a shared latent space; (2) Bidirectional Temporal Alignment, which captures temporal dependencies; and (3) Temporal Enhanced Super-Resolution, which optimizes the SR process first in latent space and then in the climate domain. Experiments using real-world datasets are conducted to evaluate the proposed method.

**Strengths:**

1. The paper addresses a gap in the existing literature, as current models fail to leverage temporal correlations in climate data super-resolution (SR). The motivation is clearly described, and the proposed solution using bidirectional temporal alignment is well-justified.
2. The paper is generally well-written and easy to understand.
3. Experiments using two real-world datasets are conducted to evaluate the proposed method.

**Weaknesses:**

1. It is unclear why, in the experiments, the low-resolution and high-resolution data are taken from different datasets. It appears that ERA5 already provides both types of data. The motivation for this choice needs clarification.
2. To comprehensively evaluate the effectiveness of the proposed method, traditional numerical simulations and statistical interpolation methods should be included as baselines.
3. As shown in Table 1, some performance improvements are very marginal. Statistical tests should be performed to determine whether these improvements are significant.
4. The authors claim that "the encoder in VQ-VAE generates discrete vectors, drawn directly from a learnable, fixed codebook, which enhances the model’s stability and interpretability". However, the paper does not provide insight or evidence regarding the model’s stability and interpretability.

**Questions:**

Same as the Weaknesses

---

### Official Review · Reviewer_xMV3 · 2025-10-30

**Soundness:** 2
**Presentation:** 2
**Contribution:** 1
**Rating:** 4
**Confidence:** 5

**Summary:**

The paper proposes a Temporal-Enhanced SR framework for climate data super-resolution that explicitly leverages temporal information. The pipeline has three parts: (a) Paired Latent Mapping uses two VQ-VAEs (LR/HR) with a shared codebook to put LR/HR fields in a unified latent space (for noise reduction and spatial consistency); (b) Bidirectional Temporal Alignment trains forward and backward alignment networks on consecutive latent frames to capture temporal dependencies; and (c) Temporal-Enhanced Super-Resolution learns a spatio-temporal SR network whose loss is regularized by forward and backward alignment to encourage time-consistent predictions. On CMIP6→ERA5 (5.625°→1.40625°), the method reports the best latitude-weighted RMSE across five key variables (Z500, T850, T2m, U10, V10), improving over VRT, SwinIR and ClimaX.

**Strengths:**

- Conceptual integration: The paper neatly combines latent-space representations with temporal regularization, aiming to handle noisy spatiotemporal climate data.

- Empirical gains: Demonstrates consistent improvements across multiple climate variables and shows that temporal information indeed benefits SR fidelity.

**Weaknesses:**

1. **Missing critical related work and baselines.**

   * The paper omits **STVD** (Spatiotemporal Video Diffusion for Precipitation Downscaling, *NeurIPS 2024*), which *explicitly* performs temporal downscaling and has been shown to outperform **VRT**, **SwinIR**, and other VSR baselines used here. STVD is a direct comparison point and a crucial omission.
   * The **CorrDiff** model (*Nature 2025*) for km-scale diffusion-based downscaling and **FNO-based neural operator methods** for arbitrary-resolution and zero-shot downscaling are both major recent baselines missing from discussion or comparison.

2. **Overstated novelty of bidirectional alignment.**

   * “Bidirectional temporal alignment” is *not novel*; it’s a core element of modern video super-resolution architectures like **BasicVSR**, **BasicVSR++**, **RVRT**, **VRT**, and **PSRT**, all of which use bidirectional or second-order temporal propagation.

3. **Overstated novelty of the paired latent mapping.**

   * The “shared codebook for LR and HR” mirrors the concept of **coupled LR/HR dictionaries** from Image Super-Resolution via Sparse Representation (Yang et al. (2010)) and codebook-prior SR in **FeMaSR**.
   * The novelty lies only in adapting this to multivariate gridded climate data, but this lineage must be acknowledged to avoid overclaiming.

4. **Incomplete discussion of alignment alternatives.**

   * The authors claim optical-flow alignment is ineffective, but omit discussing **attention-based alignment removal**, e.g., *Rethinking Alignment in VSR Transformers*, which demonstrates that strong spatiotemporal attention can replace explicit flow alignment, relevant since STVD relies on modified attention, not flow.
   * This makes the motivation for the proposed alignment regularizer weaker and under-justified.

5. **Evaluation gaps and missing metrics.**

   * The evaluation only reports **latitude-weighted RMSE**, whereas climate SR now routinely includes:

     * **Skill metrics:** Anomaly correlation coefficient (ACC).
     * **Spectral fidelity:** 1D/2D power spectra.
     * **Distributional skill:** CRPS or tail quantile errors for extremes.
     * **Structural similarity:** SSIM or gradient MSE.
   * Without these, it’s unclear whether the model preserves fine-scale variability, extremes, or spatiotemporal coherence.

6. **Potential issues in the shared latent training setup.**

   * Missing ablation:
     (a) separate vs. shared codebooks,
     (b) frozen vs. fine-tuned variants,
     (c) report codebook usage entropy and reconstruction MSEs for each VQ-VAE.

7. **Exaggerated claim about temporal alignment novelty in climate SR.**

   * The statement “there is currently no research that applies temporal alignment to climate SR” is inaccurate. STVD and other diffusion-based approaches *explicitly* incorporate temporal or sequential modeling. The correct statement would be that *explicit optical-flow-style alignment* hasn’t been explored, but temporal modeling certainly has.

**Questions:**

- Boundary handling: How are sequence edges treated (start/end of a window) when t−1 or t+1 is unavailable?
- Latent codebook interpretability: Do codewords correspond to physically meaningful motifs (e.g., jets, fronts)
- Multivariate coupling: Is ψST trained jointly across variables or with variable-specific heads?
- Spherical geometry: Are positional encodings metric-aware or plain 2D grids? Any artifacts near poles, and would periodic/rotational encodings reduce them?

---

### Meta-Review · Area_Chair_mQ5F · 2025-12-21

**Summary:**

The paper proposes a method for climate downscaling leveraging temporal structure. The reviewers agree the topic is relevant and interesting, but raise a number of concerns, including related to the evaluation procedure and comparison with related work. The authors do not provide a rebuttal. Accordingly, I must recommend rejection.

**Reviewer Concerns:**

N/A (no rebuttals)

**Reviewer Scores:**

N/A (no rebuttals)

---

### Decision · Program_Chairs · 2026-01-26

Reject